# The effects of exercise on body composition of prostate cancer patients receiving androgen deprivation therapy: An update systematic review and meta-analysis

Wenjuan Shao[1], Hanyue Zhang[2], Han Qi[1], Yimin Zhang[2]*

**1** School of Sport Science, Beijing Sport University, Beijing, China, **2** Key Laboratory of the Ministry of Education of Exercise and Physical Fitness, Beijing Sport University, Beijing, China

* ymzhangno1@163.com

**Data Availability Statement:** All relevant data are within the manuscript and its Supporting Information files.

## Abstract

Androgen deprivation therapy is a common treatment for prostate cancer. However, this therapy is associated with various adverse effects, such as increased body fat and decreased bone mineral density. Exercise may be useful for ameliorating these adverse effects, although it is not completely effective. This review aimed to clarify how exercise interventions influenced body composition and bone mineral density and to explore the most effective exercise program among prostate cancer patients who received androgen deprivation therapy. We searched the PubMed, **EMBASE**, Web of Science, EBSCO, and Cochrane Library databases for reports of randomised controlled trials that were published until October 2021. All studies involved prostate cancer patients who received androgen deprivation therapy and completed aerobic exercise, resistance exercise, and/or impact exercise training. Outcomes were defined as lean body mass, body fat mass, body fat rate, regional and whole-body bone mineral density. Thirteen reports regarding 12 randomised clinical trials (715 participants) were included. Relative to the control group, exercise intervention provided a higher lean body mass (mean difference: 0.88, 95% confidence interval: 0.40 to 1.36, P<0.01), a lower body fat mass (mean difference: -0.60, 95% confidence interval: -1.10 to -0.10, P<0.05), and a lower body fat rate (mean difference: -0.93, 95% confidence interval: -1.39 to -0.47, P<0.01). Subgroup analyses revealed greater efficacy for exercise duration of ≥6 months (vs. <6 months) and exercise immediately after the therapy (vs. delayed exercise). No significant differences were observed in the bone mineral density outcomes. Exercise can help ameliorate the adverse effects of androgen deprivation therapy in body composition, with combination exercises including resistance exercise, 8–12 repetition maximum of resistance exercise intensity, prolonged exercise duration, and performing exercise immediately after therapy providing better amelioration. And the combination of resistance and impact exercise appears to be the best mode for improving the bone mineral density.

**Funding:** The author(s) received no specific funding for this work.

**Competing interests:** The authors have declared that no competing interests exist.

## Introduction

Prostate cancer is the second most frequently diagnosed visceral cancer [1] and fifth leading cause of cancer-related mortality among men [1,2]. The GLOBOCAN estimates for 2018 include 1.3 million new prostate cancer cases and 359,000 related deaths worldwide [2], which are higher than the estimates for 2012 (1.1 million new cases and 307,500 deaths) [3] and for 2008 (899,000 new cases and 258,000 deaths) [4]. Thus, the incidence and mortality rates of prostate cancer are increasing worldwide [1–4].

Androgen deprivation therapy (ADT) is commonly used for local and metastatic prostate cancer, as androgen receptor signalling plays an important role in the survival of cancer cells [5]. One review has indicated that approximately one-third of the 2 million patients with prostate cancer in the United States are treated using ADT [6]. However, ADT is associated with various adverse effects, including increased body fat and decreases in lean body mass (LBM) and bone mineral density (BMD) [7,8]. These adverse effects lead to increased risks of obesity, sarcopenia, osteoporosis, fracture, diabetes, and cardiovascular disease [9], as well as reduced physical function and quality of life [10,11]. There is some evidence that ADT decreases LBM by an average value of 2.8% and increases the body fat rate (BFR) by an average value of 7.7% [12]. Moreover, ADT accelerates the decrease in BMD at multiple skeletal sites by 5–10-fold [13] and also increases the rate of peripheral and vertebral fractures by 4-fold [14]. Thus, secondary interventions are needed to counteract these adverse effects.

Exercise may help ameliorate the adverse effects of ADT while also providing benefits in terms of cardiopulmonary health and quality of life [15,16]. However, there is conflicting evidence regarding the effects of exercise on body composition. For example, Alberga et al. [17] and Galvao et al. [18] reported that exercise has a protective effect on LBM, although other studies indicated that not all exercise interventions significantly slowed the loss of LBM [19–23]. Alberga et al. [17] and Cormie et al. [24] reported that exercise training prevented fat accumulation, although other studies revealed no differences between the exercise intervention and control groups [18,21–23]. Several randomised controlled trials (RCTs) have also indicated that exercise did not influence local or whole-body BMD in prostate cancer patients who were treated using ADT [18,19,23], although Newton et al.'s RCT [21] and Winters-Stone et al.'s RCT [25] indicated that exercise significantly influenced the regional BMD. A cross-sectional study also indicated that self-reported exercise was positively correlated with hip BMD [26]. Therefore, further information is needed regarding the effects of exercise on body composition and BMD.

Some reviews have shown the effects of exercise on body composition and BMD. Lopez et al. [27,28] revealed that exercise had beneficial effects on body composition for prostate cancer patients receiving any treatment but it is not clear whether exercise ameliorates ADT-related adverse effects. Other reviews have described conflicting findings regarding the effects of exercise on body composition and BMD for prostate cancer patients via ADT treatment [29–32]. Three reviews showed that exercise had beneficial effects on LBM [29,31,32], but Gao et al. [30] revealed that exercise did not affect LBM. Two reviews showed that exercise had beneficial effects on body fat [30,32], but Gardner et al. [29] and Logan et al. [31] cannot obtain the same result. Logan et al. [31] revealed that exercise had a beneficial effect on BMD, but Gardner et al. [29] showed the effect of exercise on BMD was unclear. Given these conflicting findings and the dramatic increase in the number of high-quality RCTs during recent years, it is necessary to re-consider the available data. This systematic review and meta-analysis aimed to comprehensively evaluate the effects of exercise interventions on the LBM, body fat, and BMD in prostate cancer patients who received ADT. Furthermore, this study aimed to evaluate the effects of different exercise parameters (type, intensity, and duration) on LBM, body fat,

and BMD, which may be useful for establishing reference values and guiding the development of better exercise programs.

## Methods

The protocol of this systematic review and meta-analysis was registered at PROSPERO (ID: CRD42020187681) and was also published on protocol.io (dx.doi.org/10.17504/protocols.io.b4a8qshw). The protocol conformed to the Preferred Reporting Items for Systematic Reviews and Meta Analyses (PRISMA) guidelines (S1 File) [33].

### Search strategy

We searched the PubMed, EMBASE, Web of Science, EBSCO, and Cochrane Library databases for reports that were published until October 22, 2021. The search terms generally focused on 'exercise', 'training', 'physical activity', 'prostate cancer', and 'androgen deprivation therapy'. The search strategies are included in the S2 File. Manual retrieval of relevant reviews, reports, and conference abstracts was also undertaken to ensure comprehensiveness.

### Selection criteria

The inclusion criteria were based on the Population, Intervention, Comparison, Outcomes and Study framework. First, all participants were adult men diagnosed with prostate cancer, and were currently receiving ADT during the intervention, regardless of whether they had received chemotherapy, radiotherapy or other therapy. Second, the interventions included aerobic exercise, resistance exercise, and/or impact exercise. Third, the report should describe at least one relevant outcome (LBM, body fat mass [BFM], BFR, whole-body BMD, lumbar BMD, total hip BMD, and femoral neck BMD). Fourth, reports of RCTs and studies were considered appropriate if they were published in English.

The exclusion criteria were: (1) studies that did not provide ADT to all patients (unless the results were stratified according to ADT use); and (2) studies that involved interventions that combined exercise and diet, nutrition or other lifestyle.

### Data extraction and quality assessment

Data were independently extracted by two reviewers (Shao WJ and Zhang HY) using a standardised data extraction sheet, and any disputes were discussed and settled by a third person (Zhang YM). Relevant data included study-related information (first author name, publication year, study location, participant characteristics, and details of the intervention), quality assessment information, sample size and outcomes data. The outcomes of intra-group differences in the exercise and control groups were selected as priority data. Secondly, the outcomes at baseline and post-intervention were selected in the exercise and control groups, and then calculated the intra-group differences using formulas [34]. The data were presented by mean and standard deviation. If the standard deviation was not originally reported, it was calculated based on related data (e.g., quartiles and 95% confidence intervals [CIs]) using the relevant formulas [34].

The Cochrane risk of bias tool was used for the quality assessment, which assigns high, low, or unclear risks for selection bias, performance bias, detection bias, attrition bias, reporting bias, and other bias [34].

### Data analysis

The statistical analyses were performed using RevMan software (version 5.3) and Stata software (version 12.0). Absolute net differences between the intervention and control groups

were used to estimate merger effects. Outcomes were expressed as the weighted mean difference (MD) and its 95% CI. Random effect models were used given the heterogeneity of interventions. Subgroup analyses were also performed to investigate the sources of heterogeneity, which could be attributed to exercise type, exercise intensity, exercise duration, and ADT duration. The Egger's and Begg's tests were used to judge publication bias. Differences were considered statistically significant at $P$-values of $<0.05$.

## Results

### Study selection and characteristics

The search of the five databases revealed 1,749 potentially relevant reports, although 633 duplicates were removed using EndNote software. After screening the titles and abstracts, an additional 955 reports were removed and 161 full-text reports were ultimately assessed for eligibility (Fig 1).

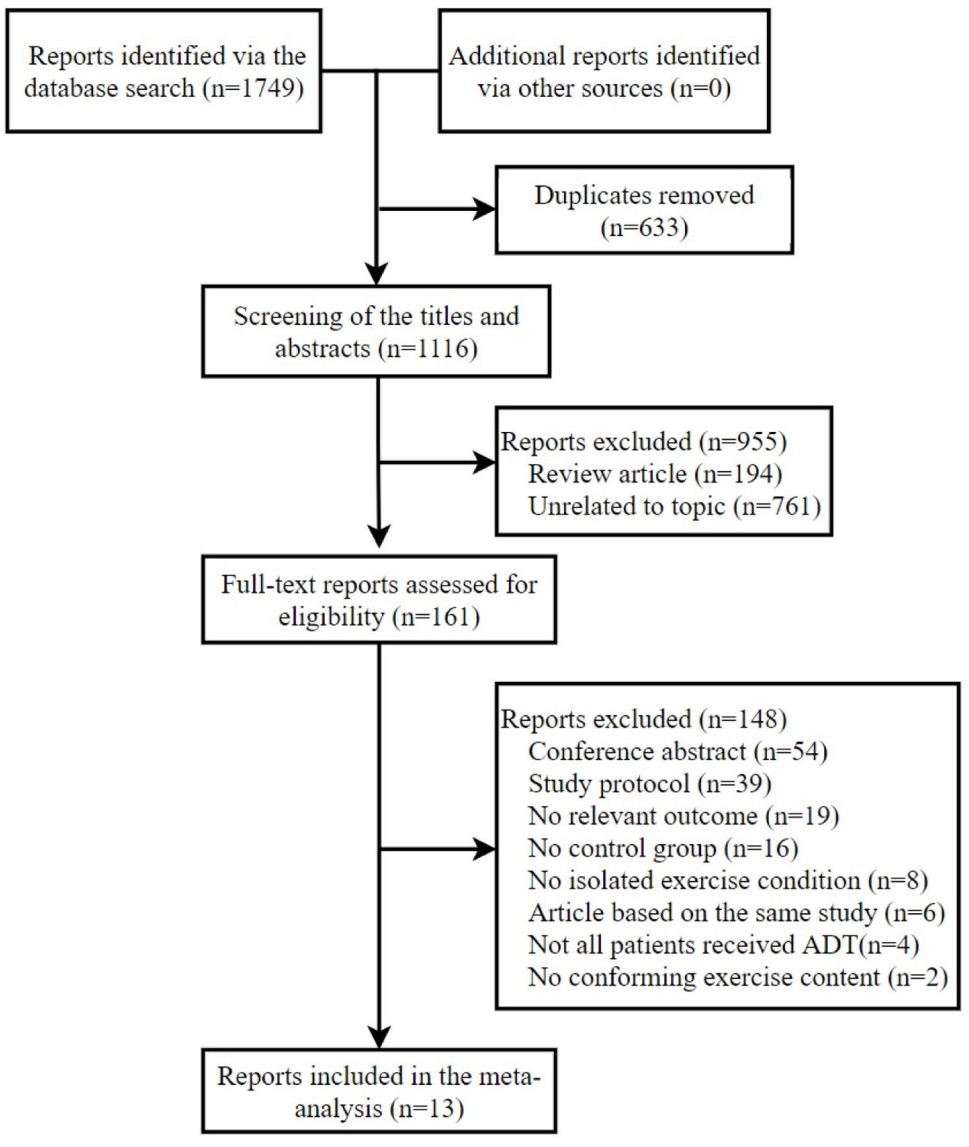

**Fig 1. PRISMA flow diagram.**

**Table 1. Characteristics of studies included in the meta-analysis.**

| Author | Year | Country | Sample size* | | Age, Mean (SD) | | Duration of ADT | Supervision |
|---|---|---|---|---|---|---|---|---|
| | | | IT | C | IT | C | | |
| **Alberga et al. [17]** | 2012 | Canada | 23 | 26 | 66.0 (—) | 66.0 (—) | —— | Supervised |
| **Cormie et al. [24]** | 2015 | Australia | 32 | 31 | 69.6 (6.5) | 67.1 (7.5) | Just started | Supervised |
| **Galvão et al. [18]** | 2010 | Australia | 29 | 28 | 69.5 (7.3) | 70.1 (7.3) | ≥2 months | Supervised |
| **Kim et al. [35]** | 2018 | Korea | 26 | 25 | 70.5 (5.0) | 71.0 (5.5) | Mean: 22 months | Home-based |
| **Lam et al. [36]** | 2020 | Australia | 13 | 12 | 69.3 (2.3) | 71.8 (1.8) | Just started | Home-based |
| **Ndjavera et al. [37]** | 2020 | UK | 24 | 26 | 71.4 (5.4) | 72.5 (4.2) | Just started | Supervised |
| **Newton et al. [21]** | 2019 | Australia | 57 | 47 | 68.7 (9.3) | 69.1 (8.4) | >2 months | Supervised |
| **Nilsen et al. [38]** | 2015 | Norway | 28 | 30 | 66.0 (6.6) | 66.0 (5.0) | ≥6 months | Supervised |
| **Taaffe et al. [22]** | 2018 | Australia | 28 | 29 | 73.0 (5.2) | 72.2 (8.4) | ≥2 months | Supervised |
| **Taaffe et al. [23]** | 2019 | Australia | 54 | 50 | 69.0 (6.3) | 67.5 (7.7) | Just started | Supervised |
| **Villumsen et al. [39]** | 2019 | Danish | 23 | 23 | 67.6 (4.9) | 69.8 (4.4) | ≥9 months | Home-based |
| **Winters-Stone et al. [25,40]** | 2014/2015 | America | 29 | 22 | 69.9 (9.3) | 70.5 (7.8) | Mean: 30 months | Supervised |

*Sample size includes lost to follow-up.

The primary analyses included 13 reports [17,18,21–25,35–40] of 12 RCTs, as two reports [25,40] described different outcomes from the same study. Data from one study were stratified based on ADT use [17]. All outcomes were assessed using dual-energy X-ray absorptiometry. The studies included 715 participants who had prostate cancer (mean age: 66–73 years), with 366 in the exercise intervention group and 349 in the control group. The studies were performed in Australia, Europe, Asia, and North America. Nine studies were performed under supervision at clinics or exercise clubs [17,18,21–25,37,38], while three [35,36,39] were performed without supervision in the patients' homes. The results were reported during 2010–2016 for six studies and during 2018–2020 for seven studies (Table 1).

All 12 studies compared the effectiveness of exercise and control interventions. Intervention duration was commonly 6 months (n = 5) [17,21–23,35] and 3 months (n = 4) [18,24,37,39], except for two 12-month studies [25,36] and one 4-month study [38]. The exercise interventions involved diverse combinations of aerobic, resistance, and/or impact exercises. Among them, three studies used resistance exercise alone [17,36,38], five studies used resistance and aerobic exercise [18,22,24,37,39], three studies used resistance and impact exercise [25,35,36], and one study used a combination of resistance, aerobic and impact exercise [23]. The control interventions included stretching exercise in two studies [25,35], usual care and suggested exercises in two [21,22], and only usual care in the remaining eight [17,18,23,24,34–39] (Table 2).

## Quality assessment and publication bias

Overall, there was a low risk of bias in included studies (Fig 2). All reports clearly described the randomisation of patients to the study groups, and eight of 12 studies (66.7%) reports allocation concealment. All studies were free of incomplete outcome data and selective reporting. However, only one of 12 studies (8.3%) involved blinding of the participants and personnel to treatment allocation, and five of 12 studies (41.7%) reported blinding of the outcome assessment. The above high risk of bias can be acceptable because it is difficult to blind an exercise intervention. Moreover, the Egger's and Begg's tests did not identify any publication bias ($P>0.05$).

**Table 2. Characteristics of the interventions and outcomes.**

| Author | Duration | Intervention group | Control group | Outcomes |
|---|---|---|---|---|
| **Alberga et al. [17]** | 6 months | Resistance exercise: 8–12 RM for 1–2 sets, 3 times per week | Usual care | (1)[↑] (3)[↓] |
| **Cormie et al. [24]** | 3 months | Aerobic exercise (70–85% $HR_{max}$ for 20–30 min) and resistance exercise (6–12 RM for 1–4 sets), twice per week for 60 min | Usual care | (1) (2)[↓] (3)[↓] (4)(5)(6) |
| **Galvão et al. [18]** | 12 weeks | Aerobic exercise (65–80% $HR_{max}$ for 15–20 min) and resistance exercise (6–12 RM for 2–4 sets), twice per week | Usual care | (1)[↑] (2)(3) |
| **Kim et al. [35]** | 6 months | Resistance exercise (8–15 RM for 2–3 sets), 2–3 times per week, plus weight-bearing exercise (11–15 RPE for 20–30 min), at least 150 min per week | Whole body stretching 3–5 times per week for 20 min | (5)(6)(7) |
| **Lam et al. [36]** | 12 months | Resistance exercise: 8–12 RM for 3 sets, 3 times per week | Usual care | (1)(2)[↓] (3)[↓](5)(7) |
| **Ndjavera et al. [37]** | 12 weeks | Aerobic exercise (55–85% $HR_{max}$, 6×5 min) and resistance exercise (10 RM for 2–4 sets), twice per week for 60 min, plus self-directed structured exercise or PA, 3 times per week for 30 min | Usual care | (1)(2) |
| **Newton et al. [21]** | 6 months | Resistance exercise (6–12 RM for 2–4 sets) and impact loading exercise (ground reaction forces 3–5× BW), twice per week for 60 min | Usual care and printed material | (1)(2)(4)(5)[↑] (6)(7)[↑] |
| **Nilsen et al. [38]** | 16 weeks | Resistance exercise (6–10 RM for 1–3 sets), 3 times per week | Usual care | (1)(2)(3)(4)(5) (6)(7) |
| **Taaffe et al. [22]** | 6 months | Aerobic exercise (70–85% $HR_{max}$ for 20–30 min) and resistance exercise (6–12 RM for 2–4 sets), twice per week for 60 min | Usual care and a booklet with PA recommendation | (1)(2) |
| **Taaffe et al. [23]** | 6 months | Aerobic exercise (60–85% $HR_{max}$ for 20–45 min), resistance exercise (6–12 RM for 2–4 sets), and impact exercise (ground reaction forces 3.4–5.2×BW), 3 times per week for 60 min | Usual care | (1)(2)(3)(4)(5) (6) |
| **Villumsen et al. [39]** | 12 weeks | Aerobic and resistance exercise, 3 times per week for 60 min | Usual care | (3) |
| **Winters-Stone et al. [25,40]** | 12 months | Resistance exercise (8–12 RM for 1–3 sets) and impact exercise (0–15% BW), 3 times per week for 60 min | Stretch and relaxation exercise | (2)[↓] (3)[↓] (5)(6) (7) |

RM: Repetition maximum, to evaluate the load intensity of resistance exercise; 1RM is defined as the maximum load; 6RM is defined as the load that repeated six times to reach the maximum load; 6RM≈R5% of 1RM; 8RM≈80% of 1RM; 12RM≈67% of 1RM.

$HR_{max}$: Maximum heart rate; RPE: Rate of perceived exertion; BW: Body weight.

[↑]increase

[↓]decrease.

(1): Lean body mass, (2): Body fat mass, (3): Body fat rate, (4): Whole-body BMD, (5): Lumbar BMD, (6): Total hip BMD, (7): Femoral neck BMD.

## Body composition

Changes in LBM were evaluated in nine studies with 562 participants. Relative to the control group, the exercise intervention significantly increased the LBM (MD: 0.88, 95% CI: 0.40 to 1.36, $P<0.01$) (Fig 3A). Changes in the BFM were evaluated in nine studies with 549 participants. Relative to the control group, the exercise intervention significantly reduced the BFM (MD: −0.60, 95% CI: −1.10 to −0.10, $P<0.05$) (Fig 3B). Changes in the BFR were evaluated in eight studies with 428 participants. Relative to the control group, the exercise intervention significantly reduced the BFR in prostate cancer patients receiving ADT (MD: −0.93, 95% CI: −1.39 to −0.47, $P<0.01$) (Fig 3C).

Subgroup analysis according to exercise type revealed that the LBM and BFR were more strongly affected by resistance exercise (LBM: MD: 1.43, 95% CI: -0.29 to 3.14, $P = 0.10$; BFR: MD: -1.48, 95% CI: -3.48 to 0.52, $P = 0.15$) than by resistance and other exercise (LBM: MD: 0.86, 95% CI: 0.16 to 1.53, $P<0.05$; BFR: MD: -1.08, 95% CI: -1.53 to -0.62, $P<0.01$), while the BFM was more strongly affected by resistance and other exercise (MD: -1.19, 95% CI: -1.99 to -0.40, $P<0.01$) than by resistance exercise solely (MD: -0.21, 95% CI: -0.85 to 0.44, $P = 0.53$). Subgroup analysis according to intensity of resistance exercise revealed that the LBM, BFM and BFR were more strongly affected by 8–12 RM (LBM: MD:2.61, 95% CI: 0.89 to 4.32,

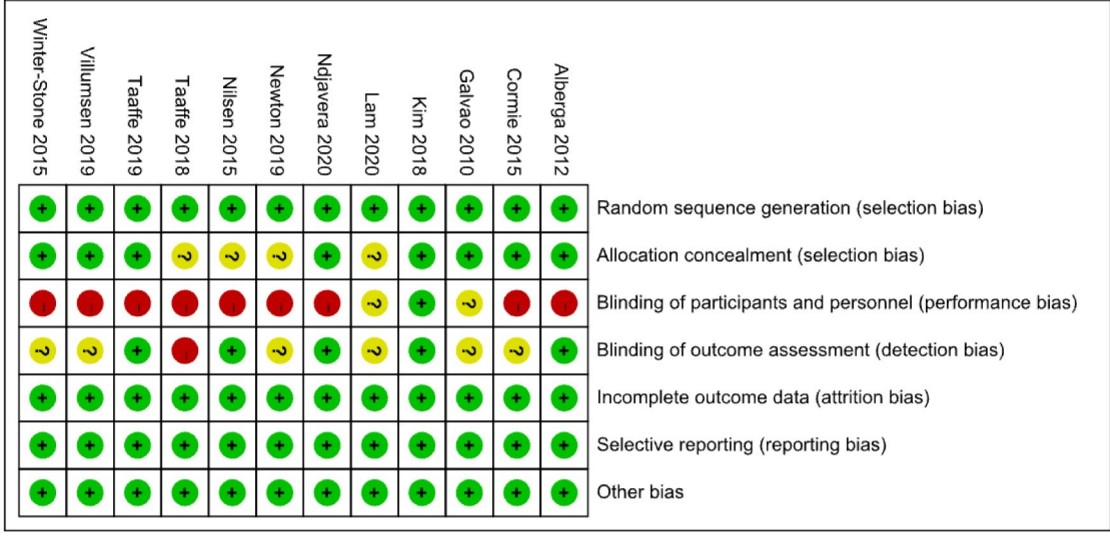

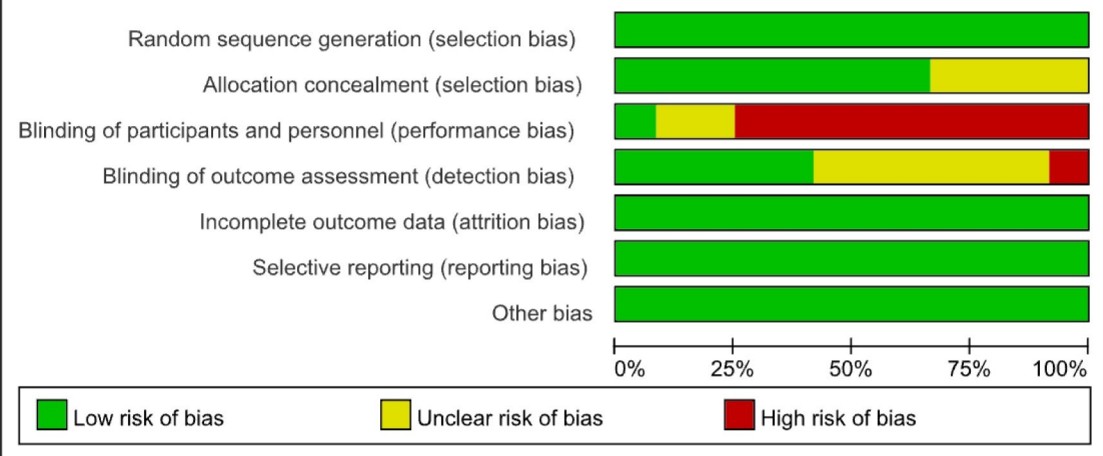

**Fig 2. Risk of bias assessment shown as percentages for each item.**

$P < 0.01$; BFM: MD: -1.69, 95% CI: -7.36 to 3.98, $P = 0.56$; BFR: MD: -2.52, 95% CI: -4.13 to -0.91, $P < 0.01$) than by 6–12 RM (LBM: MD: 0.83, 95% CI: 0.12 to 1.55, $P < 0.05$; BFM: MD: -1.15, 95% CI: -1.97 to -0.34, $P < 0.01$; BFR: MD: -1.09, 95% CI: -1.56 to -0.62, $P < 0.01$). Subgroup analysis according to exercise duration revealed that the LBM and BFR were more strongly affected by a duration of ≥6 months (LBM: MD: 1.60, 95% CI: 0.37 to 2.83, $P < 0.05$; BFR: MD: -2.01, 95% CI: -3.23 to -0.78, $P < 0.01$) than by a duration of <6 months (LBM: MD: 0.75, 95% CI: 0.23 to 1.28, $P < 0.01$; BFR: MD: -0.78, 95% CI: -1.20 to -0.36, $P < 0.01$). Subgroup analysis according to the ADT duration revealed that the BFM was more strongly affected by exercise performed immediately after ADT (MD: -1.37, 95% CI: -2.25 to -0.49, $P < 0.01$) than by exercise that was delayed after ADT (MD: -0.23, 95% CI: -0.83 to 0.38, $P = 0.47$) (Table 3).

## Bone mineral density

The whole-body BMD was evaluated in four studies with 329 participants, the lumbar BMD was evaluated in seven studies with 426 participants, the total hip BMD was evaluated in six studies with 406 participants, and the femoral neck BMD was evaluated in five studies with

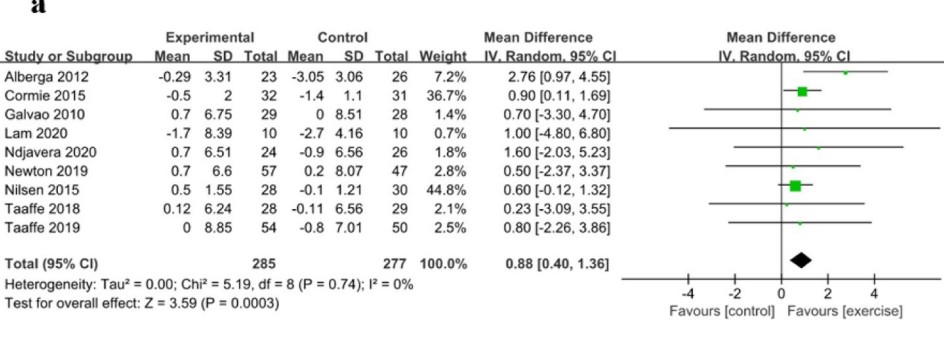

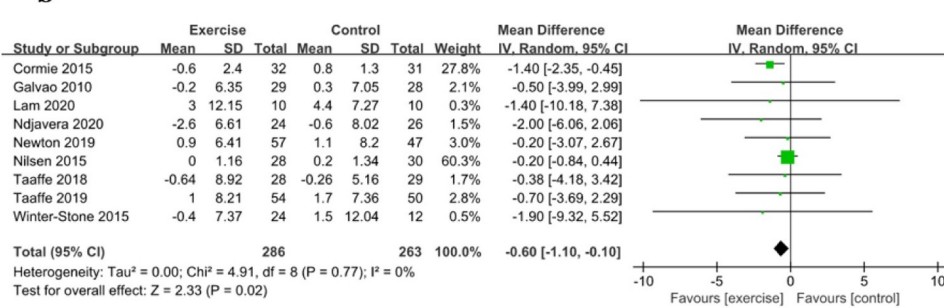

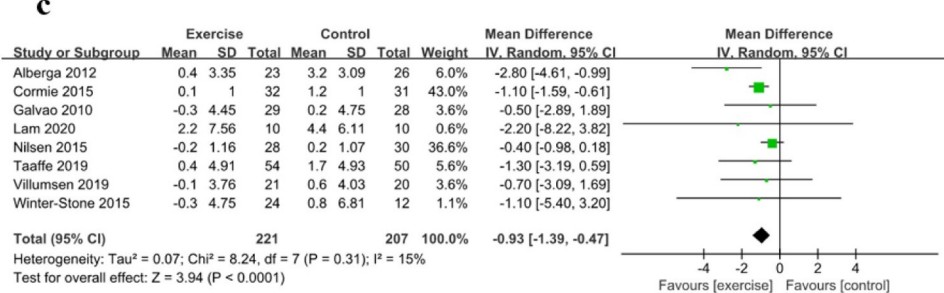

**Fig 3.** Forest Plots for (a) Lean Body Mass (LBM), (b) Body Fat Mass (BFM), and (c) Body Fat Rate (BFR).

259 participants. Relative to the control group, the exercise intervention did not significantly inhibit the loss of whole-body BMD (MD: -0.00, 95% CI: -0.01 to 0.01, *P* = 0.74), lumbar BMD (MD: 0.00, 95% CI: -0.00 to 0.01, *P* = 0.16), total hip BMD (MD: 0.00, 95% CI: -0.00 to 0.01, *P* = 0.09), and femoral neck BMD (MD: -0.00, 95% CI: -0.00 to 0.00, *P* = 0.74) (Fig 4).

## Discussion

This systematic review and meta-analysis evaluated data from recent RCTs to determine whether exercise interventions influenced the adverse effects of ADT on body composition and BMD in patients with prostate cancer. The results suggest that exercise had a beneficial effect on the LBM, BFM and BFR, but no significant effects on regional BMD and whole-body BMD. Furthermore, beneficial effects appear to be associated with combination exercises including resistance exercise, 8-12RM of resistance exercise intensity, prolonged exercise duration, and performing exercise immediately after ADT. Moreover, a combination of resistance and impact exercise appears to be the best mode for inhibiting the loss of BMD. These findings are valuable for guiding the exercise of prostate cancer patients who are receiving ADT.

**Table 3. Subgroup analysis of the effects of exercise on body composition.**

| | No. of studies | Sample size | | I² | MD (95% CI) | P value |
|---|---|---|---|---|---|---|
| | | IT | C | | | |
| **Type of exercise** | | | | | | |
| **LBM** | | | | | | |
| Resistance exercise | 3 | 61 | 66 | 58% | 1.43 [-0.29, 3.14] | 0.10 |
| Resistance and other exercise | 6 | 224 | 211 | 0% | 0.86 [0.16, 1.56] | <0.05 |
| **BFM** | | | | | | |
| Resistance exercise | 2 | 38 | 40 | 0% | -0.21 [-0.85, 0.44] | 0.53 |
| Resistance and other exercise | 7 | 248 | 223 | 0% | -1.19 [-1.99, -0.40] | <0.01 |
| **BFR** | | | | | | |
| Resistance exercise | 3 | 61 | 66 | 69% | -1.48 [-3.48, 0.52] | 0.15 |
| Resistance and other exercise | 5 | 160 | 141 | 0% | -1.08 [-1.53, -0.62] | <0.01 |
| **Intensity of resistance exercise*** | | | | | | |
| **LBM** | | | | | | |
| 8–12 RM | 2 | 33 | 36 | 0% | 2.61 [0.89, 4.32] | <0.01 |
| 6–12 RM | 5 | 200 | 185 | 0% | 0.83 [0.12, 1.55] | <0.05 |
| **BFM** | | | | | | |
| 8–12 RM | 2 | 34 | 22 | 0% | -1.69 [-7.36, 3.98] | 0.56 |
| 6–12 RM | 5 | 200 | 185 | 0% | -1.15 [-1.97, -0.34] | <0.01 |
| **BFR** | | | | | | |
| 8–12 RM | 3 | 57 | 48 | 0% | -2.52 [-4.13, -0.91] | <0.01 |
| 6–12 RM | 3 | 115 | 109 | 0% | -1.09 [-1.56, -0.62] | <0.01 |
| **Duration of exercise** | | | | | | |
| **LBM** | | | | | | |
| <6 months | 4 | 113 | 115 | 0% | 0.75 [0.23, 1.28] | <0.01 |
| ≥6 months | 5 | 172 | 162 | 0% | 1.60 [0.37, 2.83] | <0.05 |
| **BFM** | | | | | | |
| <6 months | 4 | 113 | 115 | 36% | -0.75 [-1.60, 0.09] | 0.08 |
| ≥6 months | 5 | 173 | 148 | 0% | -0.54 [-2.28, 1.19] | 0.54 |
| **BFR** | | | | | | |
| <6 months | 4 | 110 | 109 | 10% | -0.78 [-1.20, -0.36] | <0.01 |
| ≥6 months | 4 | 111 | 98 | 0% | -2.01 [-3.23, -0.78] | <0.01 |
| **Duration of ADT** | | | | | | |
| **LBM** | | | | | | |
| Immediate exercise after ADT | 4 | 120 | 117 | 0% | 0.93 [0.18, 1.67] | <0.05 |
| Delayed exercise after ADT | 5 | 165 | 160 | 20% | 1.02 [0.08, 1.96] | <0.05 |
| **BFM** | | | | | | |
| Immediate exercise after ADT | 4 | 120 | 117 | 0% | -1.37 [-2.25, -0.49] | <0.01 |
| Delayed exercise after ADT | 5 | 166 | 146 | 0% | -0.23 [-0.83, 0.38] | 0.47 |
| **BFR** | | | | | | |
| Immediate exercise after ADT | 3 | 96 | 91 | 0% | -1.12 [-1.60, -0.64] | <0.01 |
| Delayed exercise after ADT | 5 | 125 | 116 | 35% | -0.97 [-1.97, 0.04] | 0.06 |

* RCTs would be exclude if the intensity of resistance exercise is not 8–12 RM or 6–12 RM.

## Body composition

Exercise could ameliorate the adverse effects on body composition of ADT. For exercise type, resistance exercise is the most effective way to increase LBM and decrease BFR. Subgroup

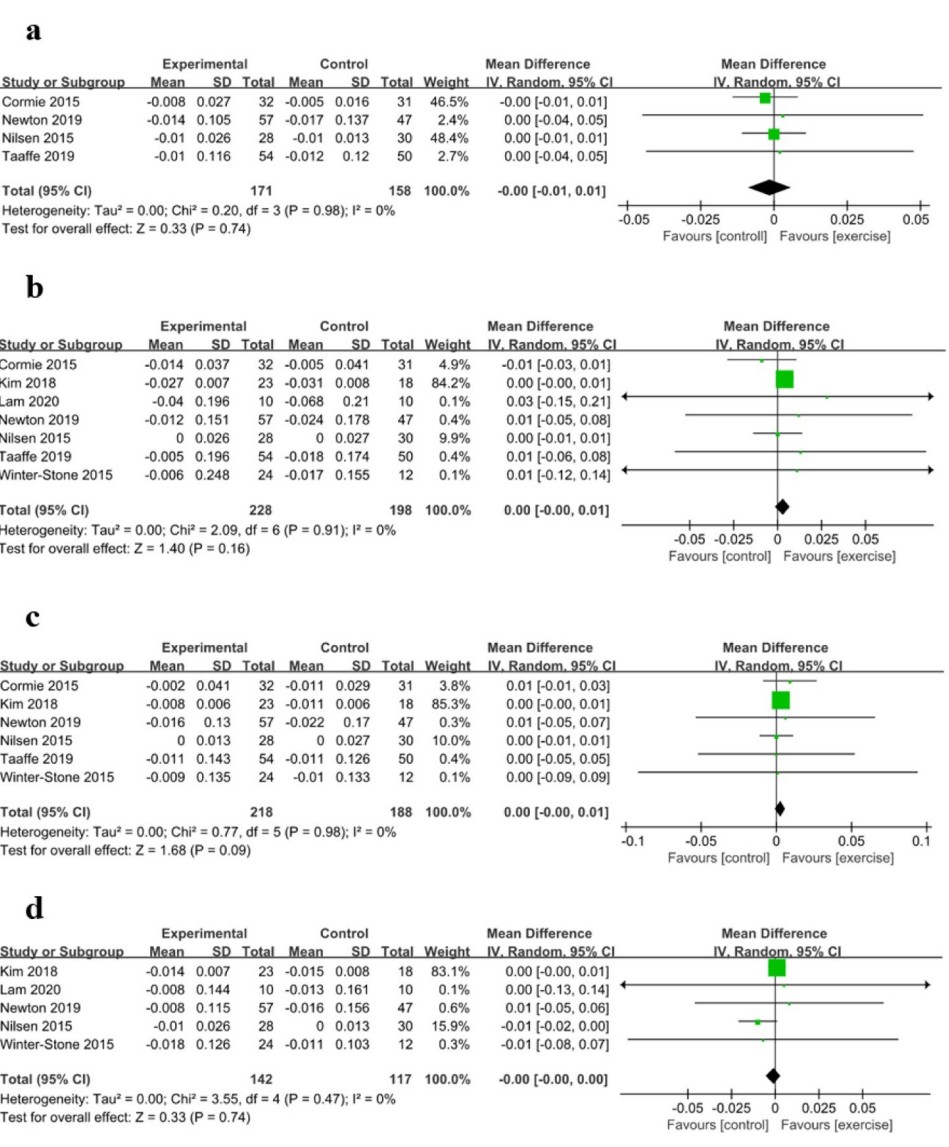

**Fig 4.** Forest Plots for (a) Whole-body Bone Mineral Density (BMD), (b) Lumbar BMD, (c) Total Hip BMD, and (d) Femoral Neck BMD.

analysis of this review showed that resistance exercise had higher effect sizes on LBM and BFR. And a three-arm RCT (74 prostate cancer patients receiving ADT) also revealed that, relative to the control group, resistance exercise but not aerobic exercise significantly ameliorated LBM and BFR [17]. That's probably because resistance exercise stimulates the mTORC1 signaling pathway, which is responsible for muscle anabolism [31]. It could also be because resistance exercise increased human growth hormone and dehydroepiandrosterone, which play an important role in maintaining and increasing LBM, in prostate cancer patients receiving ADT [41]. With the synthesis of muscle and the increase of LBM, the BFR naturally decreases. However, resistance exercise alone is not the best exercise type to decrease BFM. Subgroup analyses of this review showed that resistance combined with other exercises had a higher effect size on BFM. The difference of effect sizes between two subgroups on BFM was large than that on LBM and BFR. At the same time, aerobic exercise or impact exercise cannot reduce the effects

of resistance exercise, and aerobic exercise may even promote a greater effect of resistance exercise on muscle mass [31], so combination exercises including resistance exercise would be better without considering other factors such as exercise intensity and exercise volume.

For resistance exercise intensity, the intensity of 8–12 RM appeared to be more effective. Most of our included RCTs used resistance exercise intensity at 6–12 or 8–12 RM. Among these RCTs, the results of exercise intensity of 6–12 RM on body composition were not inconsistent and most were not significant, while the results of exercise intensity of 8–12 RM were consistent and significant. Subgroup analyses also showed that a lower resistance exercise intensity (8–12 RM) had higher effect sizes on LBM, BFM and BFR. 6–12 RM (67%-85% of 1RM) is moderate-to-high intensity, 8–12 RM (67%-80% of 1RM) is moderate intensity, and 6-12RM contains 8–12 RM. This is demonstrated that moderate intensity exercise is better to achieve significant benefits than moderate-to-high intensity exercise in patients with prostate cancer. In addition, low-intensity resistance exercise is also an important factor to be considered. There were evidences suggested that resistance exercise can increase muscle mass and strength in men, and the increased effect between low-intensity and high-intensity exercise has no significant difference [42,43]. The study of Lopez et al. also suggested that the low-to-high intensity resistance exercise is as effective as moderate-to-high intensity for enhancing body composition in prostate cancer patients [27]. Although it is difficult to assess and quantify the effect of low-intensity exercise on body composition due to the lack of reporting, low-intensity exercise may be better given the above description and the patient's age, physical condition and exercise compliance. Thus, the low-to-moderate resistance exercise would be recommended in the future.

For exercise duration, subgroup analyses revealed that it influenced the effects of the intervention, with longer duration (≥6 months vs. <6 months) exerting greater effects on LBM and BFR. The meta-analysis of Gao et al. [30] also reported that, compared with the duration in 6 months, the exercise duration over 6 months had a higher effect size on BMI of patients with prostate cancer. A 12-month RCT [36], which measuring outcomes at 6 weeks and 6 and 12 months, suggested that exercise can alleviate ADT-related body fat at 12 months, but with no significant difference at 6 weeks and 6 months between the exercise group and the control group. So longer exercise duration may be more effective in improving the adverse effects of ADT.

The duration of ADT is also a limiting factor in the effect of intervention. Subgroup analyses did not reveal the effect of exercise duration on LBM and BFR because the closely MDs of LBM (0.93 vs. 1.02) and BFR (-1.12 vs. -0.97) in the two subgroups, but it revealed the effect of exercise duration on BFM. Exercise immediately after ADT had greater effect on BFM than delayed exercise after ADT. That's probably because the ADT-related adverse effects on body composition emerging during the initial months of ADT [8,44], and exercise immediately could mitigate the trend. Taaffe et al. [22] explored whether ADT duration influenced the response to exercise in patients who had already begun ADT, and revealed that long-term ADT (18 months vs. 6 months) was associated with more favourable response to exercise than short-term ADT in terms of body composition. The mechanism underlying this difference was not attributed to lower initial values and more improved room in the long-term group. Thus, it is preferable to start exercise and ADT at the same time, and further research would be exploded the optimal timing in prostate cancer patients who have already begun ADT.

Therefore, although the effects of exercise on body composition are complex and multifaceted, it appears likely that choosing resistance exercise combined with other exercise, choosing 8–12 RM as the resistance exercise intensity, extending the duration of the exercise intervention, and performing exercise and ADT at the same time will be effective.

## Bone mineral density

Decreasing BMD is an important adverse effect of ADT, as it increases the risks of osteoporosis and fractures [7,45,46] and seriously affects the patient's functional independence and quality of life [11]. This study failed to detect exercise-related effects on regional and whole-body BMD. Among the studies we included, two of seven studies revealed significant differences in BMD between the exercise and control groups [21,25]. Newton et al. [21] reported that a 6-month resistance and impact exercise had significant effects on the lumbar BMD and femoral neck BMD. Winters-Stone et al. [25] reported that a 12-month resistance and impact exercise intervention had a slight protective effect on the lumbar ($L_4$) BMD but not the hip BMD. Both the two studies used different exercise duration, exercise frequency and exercise intensity, but the same exercise type, indicating that resistance exercise combined with impact exercise might be an effective exercise intervention type to protect BMD. In fact, this exercise type is known for its osteogenic capacity, and its positive effect on BMD has also been demonstrated in older healthy men and older patients with breast cancer [47,48].

Besides this exercise type, can other exercise types have a significant effect on BMD of P prostate cancer patients during ADT? (1) Resistance and aerobic exercise: A three-arm RCT performed by Newton et al. [21] revealed that, under the same intervention conditions, the resistance and impact exercise, but not resistance and aerobic exercise, had a significant effect on BMD. This is because impact exercise loads the bones in different ways, relative to aerobic exercise, and promotes greater osteogenesis [49]. (2) Resistance, aerobic and impact exercise: Taaffe et al. [23] reported that BMD did not improve by a combination of aerobic, resistance, and impact exercise but a significant influence was obtained by the identical exercise duration and similar exercise intensity than in the study of Newton et al. [21]. Aerobic exercise also does not appear to reduce the combined effects of resistance and impact exercise on BMD [49,50]. The findings of Taaffe et al. [23] might be related to the high proportion of aerobic exercise, which may result in insufficient amounts of resistance and impact exercise to achieve significant results. (3) Football exercise: Uth et al. [20] reported that 32 weeks of football exercise significantly influenced the lower limbs BMD such as hip BMD and femoral shaft BMD. The effects might be related to variable impact forces acting on the bones from different angles in football exercise, with intermittent accelerations and decelerations. And there was evidence suggested that mechanical force diversity and dynamics during exercise contribute to increased bone mass [51]. Thus, football exercise can also effectively protect the BMD of patients. However, Uth et al. [20] reported five injuries caused by football in fewer than 30 patients, and no adverse events were reported by Newton et al. [21] and Winters-Stone et al. [25]. From the safety and efficacy perspective, a combination of resistance and impact exercise appears to provide the best clinical value for inhibiting BMD loss in prostate cancer patients receiving ADT.

## Limitations

One limitation of our meta-analysis is that the sample sources in the included studies are diverse. Some RCTs recruited patients who had received ADT for several months [18,21,39], and some recruited patients who had never received ADT [23,37]. The RCT of Nilsen et al. [38] recruited moderate or high risk prostate cancer patients who had received high-dose radiotherapy. And the RCT of Taaffe et al. [22] recruited patients who had participated in other clinical trial. So there would be a bias because the samples have received different treatments. Another limitation is that the mode of exercise is not comprehensive. Our meta-analysis only involved the exercise interventions which were multiple combinations of aerobic, resistance, and/or impact exercises. The studies of football exercise were excluded [19,20] in

order to standardize the intervention and quantify the content of exercise, and to explore the appropriate volume and intensity of exercise.

## Conclusion

This review suggested that exercise ameliorated the ADT-related side effects on body composition among prostate cancer patients, with longer and 8-12RM resistance exercise providing better amelioration. Starting exercise and ADT at the same time could also providing better amelioration. Moreover, this review found no direct evidence that exercise ameliorated the ADT-related decrease in BMD, which requires further study. Future studies should focus more closely on resistance and impact exercise.

## Supporting information

**S1 File. PRISMA-P 2015 checklist.**
(DOCX)

**S2 File. Search strategies for databases.**
(DOCX)

## Acknowledgments

We thank the authors of the studies included in this meta-analysis.

## Author Contributions

**Conceptualization:** Wenjuan Shao, Hanyue Zhang, Han Qi, Yimin Zhang.

**Data curation:** Wenjuan Shao, Hanyue Zhang.

**Formal analysis:** Wenjuan Shao, Hanyue Zhang, Han Qi, Yimin Zhang.

**Methodology:** Wenjuan Shao, Hanyue Zhang, Han Qi, Yimin Zhang.

**Software:** Wenjuan Shao, Han Qi.

**Writing – original draft:** Wenjuan Shao.

**Writing – review & editing:** Wenjuan Shao, Yimin Zhang.

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
