## [Decision Letter · Decision Letter 0]

19 Dec 2021

PONE-D-21-35409The effects of exercise on body composition of prostate cancer patients receiving androgen deprivation therapy: an update systematic review and meta-analysisPLOS ONE

Dear Dr. Zhang,

Thank you for submitting your manuscript to PLOS ONE. After careful consideration, we feel that it has merit but does not fully meet PLOS ONE’s publication criteria as it currently stands. Therefore, we invite you to submit a revised version of the manuscript that addresses the points raised during the review process. The revision requests for this particular manuscript is acknowledge to be significant but it is believed that if the authors are able to meet these revision requests, it will significantly improve this manuscript. 

We look forward to receiving your revised manuscript.

Kind regards,

Henry Woo

Academic Editor

PLOS ONE

Journal Requirements:

Reviewers' comments:

Reviewer's Responses to Questions

**Comments to the Author**

1. Is the manuscript technically sound, and do the data support the conclusions?

Reviewer #1: Yes

Reviewer #2: Partly

2. Has the statistical analysis been performed appropriately and rigorously? 

Reviewer #1: I Don't Know

Reviewer #2: No

3. Have the authors made all data underlying the findings in their manuscript fully available?

Reviewer #1: Yes

Reviewer #2: Yes

4. Is the manuscript presented in an intelligible fashion and written in standard English?

Reviewer #1: Yes

Reviewer #2: Yes

5. Review Comments to the Author

Reviewer #1: The selection of trials appears to be apprioriate.

The conclusions appear reasonable.

A few comments

1. Some of the text refers to the fact that ADT causes a decline in bone mineral density and exercise may "ameliorate" that but I would expect more of a focus in the results and discussion about "lack of decline" in BMD rather than an "improvement" I could not see that this was addressed in the manuscript. This sentence in the results could be modified "Relative to the control group, the exercise intervention was not associated with significant improvements in the whole-body BMD". This led me to think that the investigators were searching for an improvement... rather than lack of a decline...

2. In general I would prefer a change in some of the language used throughout the manuscript.

particularly in relation to the use of the term "pca patients". I think it is best to switch that term around, remove the "pca" abbreviation. Focus on the patient and use the term "patients with prostate cancer" instead.

In addition I feel that we have moved on from using the word "elderly" in publications now in preference to the term "older" and "older adults" with prostate cancer. A minor point perhaps but I feel that the language matters in such publications.

3. Please help the reader and define RM somewhere to enable enhanced understanding when terms like "8-12RM" are used.

4. What does "starting exercise after ADT" mean? Is that with the ADT? Is that before short term ADT is completed? Can this be clarified in the text please? This line in the conclusion could also be clarified "The best time to start exercise is immediately after ADT"

5.Meta analyses using individual patient data can yield more robust results. I imagine this analysis is not suited to technique that given the heterogeneity of the interventions?

6. What is a guarantor?

7. What is the meaning of this sentence in the discussion "It is difficult to explore the probably reasons

for the significant differences, but a lower exercise intensity (8-12 RM) will be better, considering the effect of exercise and the age and physical condition of Pca patients"?

8. And the next sentence needs rewording too "while several evidences suggested" Page 20 line 294

Reviewer #2: I commend the authors for their efforts to put this comprehensive SRMA in prostate cancer, including a range of body composition components following exercise in this population. However, I do not think this study is up-to-date or brings novelty to this population.

Although the authors justify their study by saying that “Three reviews have described conflicting findings regarding the effects of exercise for PCa patients who were treated using ADT”, several SRMAs were recently published. The angle proposed by the authors is not expanding knowledge on the topic. Please, find three recent SRMA that have investigated body composition:

Bigaran et al. The effect of exercise training on cardiometabolic health in men with prostate cancer receiving androgen deprivation therapy: a systematic review and meta-analysis. Prostate Cancer Prostatic Dis. 2021 Mar;24(1):35-48. doi: 10.1038/s41391-020-00273-5. Epub 2020 Aug 28. PMID: 32860010.

Lopez et al. Resistance Exercise Dosage in Men with Prostate Cancer: Systematic Review, Meta-analysis, and Meta-regression. Med Sci Sports Exerc. 2021 Mar 1;53(3):459-469. doi: 10.1249/MSS.0000000000002503. PMID: 32890199; PMCID: PMC7886340.

Lopez et al. Interventions for improving body composition in men with prostate cancer: A systematic review and network meta-analysis. Med Sci Sports Exerc. Published Ahead-of-Print.

Additionally, the meta-analysis model utilised is another critical issue. First, a fixed-effect model is not feasible in this setting, even with low heterogeneity. A random-effects model is highly recommended because of the high variability in this setting such as population characteristics, exercise modalities utilised, and outcomes. Second, several issues were identified in data extraction, such as the number utilised for meta-analyses and those reported within studies. Find a table below concerning the results on lean body mass:

Reported Shao et al., 2021

Study Mean difference 95% CI Mean difference 95% CI

Alberga et al., 2012 2.76 0.87 to 4.65 2.76 0.97 to 4.55

Cormie et al., 2015 0.7 -0.1 to 1.6 0.90 0.11 to 1.69

Galvao et al., 2010 0.8 0.0 to 1.5 0.70 -3.30 to 4.70

Lam et al., 2020 Not reported Not reported 1.00 -4.80 to 6.80

Ndjavera et al., 2020 1.2 -1.2 to 3.7 1.60 -2.03 to 5.23

Newton et al., 2019 0.5

0.5 -3.6 to 4.6

-4.0 to 5.0 0.50 -2.37 to 3.37

Nilsen et al., 2015 0.5 -0.2 to 1.2 0.60 -0.12 to 1.32

Taaffe et al., 2018 0.3 -0.3 to 0.8 0.23 -3.09 to 3.55

Taaffe et al., 2019 0.8 -3.5 to 5.1 0.80 -2.26 to 3.86

This issue was also observed in the other outcomes investigated by the authors.

Key concerns:-

 1) the rationale for the study is missing key points and is not considering the most up-to-date literature, and 2) issues were identified in data extraction and affected meta-analysis results.

6. PLOS authors have the option to publish the peer review history of their article (what does this mean?). If published, this will include your full peer review and any attached files.

Reviewer #1: **Yes: **Christopher Steer

Reviewer #2: No

---

## [Author Response · Author response to Decision Letter 0]

27 Jan 2022

Journal Requirements:

Answer: We have restyled the manuscript according to the PLOS ONE style templates.

Answer: We have attached the captions of the Supporting information files (line 554-556 of the ‘Manuscript’). We have updated any in-text citations accordingly (line 100, line 107).

Reviewer #1: The selection of trials appears to be apprioriate.

The conclusions appear reasonable.

The comments and answers:

1. Some of the text refers to the fact that ADT causes a decline in bone mineral density and exercise may "ameliorate" that but I would expect more of a focus in the results and discussion about "lack of decline" in BMD rather than an "improvement" I could not see that this was addressed in the manuscript. This sentence in the results could be modified "Relative to the control group, the exercise intervention was not associated with significant improvements in the whole-body BMD". This led me to think that the investigators were searching for an improvement... rather than lack of a decline...

Answer: We apologize for the lack of a clear distinction between “lack of decline” and “improvement” in BMD. In the results and discussion section of the manuscript, we modified “ameliorate BMD” and “improve BMD” to “inhibit BMD loss” or “protect BMD”. This sentence also be modified to “Relative to the control group, the exercise intervention did not significantly inhibit the loss of whole-body BMD” (line 244-245 of the ‘Manuscript’).

2. In general I would prefer a change in some of the language used throughout the manuscript.

particularly in relation to the use of the term "pca patients". I think it is best to switch that term around, remove the "pca" abbreviation. Focus on the patient and use the term "patients with prostate cancer" instead.

In addition I feel that we have moved on from using the word "elderly" in publications now in preference to the term "older" and "older adults" with prostate cancer. A minor point perhaps but I feel that the language matters in such publications.

Answer: In the revised manuscript, the abbreviation “PCa” is not used and the word “elderly” is replaced by “older”.

3. Please help the reader and define RM somewhere to enable enhanced understanding when terms like "8-12RM" are used.

Answer: We apologize for not being clear on this. RM (repetition maximum) can be used to evaluate the load intensity of resistance exercise; 1RM is defined as the maximum load; 6RM is defined as the load that repeated six times to reach the maximum load; 8RM is defined as the load that repeated eight times to reach the maximum load. 6RM≈85% of 1RM; 8RM≈80% of 1RM; 12RM≈67% of 1RM. 

In the revised manuscript, we have added the definition of RM and the explanation of related terms such as 6RM to the notes of Table 2, where the terms appeared for the first time (line 184-186).

4. What does "starting exercise after ADT" mean? Is that with the ADT? Is that before short term ADT is completed? Can this be clarified in the text please? This line in the conclusion could also be clarified "The best time to start exercise is immediately after ADT"

Answer: We apologize for not being clear on this. The population included in this study was prostate cancer patients receiving ADT. Some of them begin to exercise after receiving ADT a few months (at least ≥2 months), described in the text as “delayed exercise after ADT” (Table 3). Others begin to exercise immediately after receiving ADT, meaning that ADT and exercise begin at the same time, described as “immediate exercise after ADT” (Table 3).

In the revised manuscript, we modified “start exercise immediately after ADT” in line 322 and 327 to “start exercise and ADT at the same time”. We also modified the sentence “The best time to start exercise is immediately after ADT” in the conclusion to “Starting exercise and ADT at the same time could also providing better amelioration” (line 383). 

5. Meta analyses using individual patient data can yield more robust results. I imagine this analysis is not suited to technique that given the heterogeneity of the interventions?

Answer: I strongly agree that meta-analyses using individual patient data can yield more robust results, but it is very hard to obtain the data of each patient. This study used summary data from the original studies for a meta-analysis, and can also yield valuable results. In fact, the vast majority of published meta-analyses use summary data from the original data. 

The heterogeneity of the interventions does exist. In order to reduce heterogeneity, we have formulated rigorous selection criteria in terms of population characteristics and intervention types. The types of intervention included only aerobic, resistance or impact exercise, and any other types of intervention such as football were excluded. We also abandoned the fixed effect model and selected the random effect model. I think this study is suitable for meta-analysis technology, not only because we control the heterogeneity as much as possible, but also because there are similar studies using this technology. 

(e.g. Ussing, Anja et al. “Supervised exercise therapy compared with no exercise therapy to reverse debilitating effects of androgen deprivation therapy in patients with prostate cancer: a systematic review and meta-analysis.” Prostate cancer and prostatic diseases, 10.1038/s41391-021-00450-0.)

6. What is a guarantor?

Answer: The guarantor is one of the principal authors who takes final accountability for the integrity of the entire work. According to an article published in PLoS One in 2019 (Title: Scientific misconduct and accountability in teams; doi: 10.1371/journal.pone.0215962.), guarantors made substantial contributions to the research but also made efforts to verify and uphold the integrity of the study. And the first author is usually suggested to be the guarantor.

In this study, all the authors recommend the first author WS as the guarantor. We have modified it in the “Author Contributions” section.

7. What is the meaning of this sentence in the discussion "It is difficult to explore the probably reasons

for the significant differences, but a lower exercise intensity (8-12 RM) will be better, considering the effect of exercise and the age and physical condition of Pca patients"?

Answer: This sentence has been modified to “Although it is difficult to assess and quantify the effect of low-intensity exercise on body composition due to the lack of reporting, low-intensity exercise may be better given the above description and the patient's age, physical condition and exercise compliance” in page 21 line 297-300. 

8. And the next sentence needs rewording too "while several evidences suggested" Page 20 line 294

Answer: This sentence has been modified to “There were evidences suggested that resistance exercise can increase muscle mass and strength in men, and the increased effect between low-intensity and high-intensity exercise has no significant difference” in page 21 line 292-294.

Reviewer #2: 

The first key concern and answer: 

I commend the authors for their efforts to put this comprehensive SRMA in prostate cancer, including a range of body composition components following exercise in this population. However, I do not think this study is up-to-date or brings novelty to this population.

Although the authors justify their study by saying that “Three reviews have described conflicting findings regarding the effects of exercise for PCa patients who were treated using ADT”, several SRMAs were recently published. The angle proposed by the authors is not expanding knowledge on the topic. Please, find three recent SRMA that have investigated body composition:

Bigaran et al. The effect of exercise training on cardiometabolic health in men with prostate cancer receiving androgen deprivation therapy: a systematic review and meta-analysis. Prostate Cancer Prostatic Dis. 2021 Mar;24(1):35-48. doi: 10.1038/s41391-020-00273-5. Epub 2020 Aug 28. PMID: 32860010.

Lopez et al. Resistance Exercise Dosage in Men with Prostate Cancer: Systematic Review, Meta-analysis, and Meta-regression. Med Sci Sports Exerc. 2021 Mar 1;53(3):459-469. doi: 10.1249/MSS.0000000000002503. PMID: 32890199; PMCID: PMC7886340.

Lopez et al. Interventions for improving body composition in men with prostate cancer: A systematic review and network meta-analysis. Med Sci Sports Exerc. Published Ahead-of-Print.

1) Key concern: the rationale for the study is missing key points and is not considering the most up-to-date literature.

Answer: We thank you for providing the three recent SRMA. We revised the introduction and discussion section of the manuscript refer to the three SRMA.

The first SRMA (Bigaran et al. 2021) showed positive effects of exercise on body composition in prostate cancer receiving ADT. The second SRMA (Lopez et al. 2021) described the effect of resistance exercise on body composition of patients with prostate cancer, and focused on the dosage of exercise. The third SRMA’s (Lopez et al. 2021) purpose was “to investigate the most effective intervention for improving body composition outcomes in prostate cancer patients during or following treatment”. Compared with the three SRMA, the present study has the following characteristics.

(1) The SRMA by Bigaran et al. obtained the same results on body composition as the present study. However, compared with the SRMA of Bigaran et al., the present study included more original studies (such as Alberga et al. 2012, Winters-Stone et al.2015, Taaffe et al. 2019, Villumsen et al.2019, Ndjavera et al. 2020, and Lam et al. 2020) and more samples, which provided more sufficient evidence.

(2) The literature search of the 2nd and 3rd SRMA by Lopez et al. was up to November 2019 and December 2020 respectively. While the literature search of the present study was up to October 2021. New RCT (Lam et al. 2020) was included in the present study. 

(3) The participants included in the 2nd and 3rd SRMA were prostate cancer patients receiving any treatment. While the present study only reported prostate cancer patients receiving ADT and performed subgroup analyses according to ADT duration.

(4) For intervention type. The interventions in the 3rd SRMA included exercise/physical activity and/or nutrition, while the interventions in the present included aerobic exercise, resistance exercise, and/or impact exercise. The type of intervention in the present study was more specific.

(5) For exercise dosage. The 2nd SRMA, which described the volume of resistance exercise, showed that “lower volume at moderate to high intensity is as effective as higher volume for enhancing body composition”. While the present study described the intensity of resistance exercise, and showed that moderate intensity (8-12 RM) is more effective than moderate to high intensity (6-12 RM) on body composition.

(6) None of the 3 SRMA described the BMD outcomes, as the present study did.

Thus, the perspective proposed of the present study can expand knowledge to a certain extent.

The second key concern and answer: 

Additionally, the meta-analysis model utilised is another critical issue. First, a fixed-effect model is not feasible in this setting, even with low heterogeneity. A random-effects model is highly recommended because of the high variability in this setting such as population characteristics, exercise modalities utilised, and outcomes. Second, several issues were identified in data extraction, such as the number utilised for meta-analyses and those reported within studies. Find a table below concerning the results on lean body mass:

Reported Shao et al., 2021

Study Mean difference 95% CI Mean difference 95% CI

Alberga et al., 2012 2.76 0.87 to 4.65 2.76 0.97 to 4.55

Cormie et al., 2015 0.7 -0.1 to 1.6 0.90 0.11 to 1.69

Galvao et al., 2010 0.8 0.0 to 1.5 0.70 -3.30 to 4.70

Lam et al., 2020 Not reported Not reported 1.00 -4.80 to 6.80

Ndjavera et al., 2020 1.2 -1.2 to 3.7 1.60 -2.03 to 5.23

Newton et al., 2019 0.5

0.5 -3.6 to 4.6

-4.0 to 5.0 0.50 -2.37 to 3.37

Nilsen et al., 2015 0.5 -0.2 to 1.2 0.60 -0.12 to 1.32

Taaffe et al., 2018 0.3 -0.3 to 0.8 0.23 -3.09 to 3.55

Taaffe et al., 2019 0.8 -3.5 to 5.1 0.80 -2.26 to 3.86

This issue was also observed in the other outcomes investigated by the authors.

2) Key concern: issues were identified in data extraction and affected meta-analysis results.

Answer: We thank you for the comment. We accepted the suggestion to use the random effect model. We modified the text content, numbers and figures in the manuscript accordingly such as Table 3 and the part of “Body composition” in the “results” and “discussion”. Although some numbers and the results of some subgroup analyses have changed, the conclusions of the present study have not changed after using the random effect model. 

In addition, we apologize for the misunderstanding caused by the unclear description of the data extraction content and priority order. There were no issues in data extraction in the present study. We modified the “Data extraction” section in the manuscript for clarity. Then, I would introduce the process of data extraction, entry and effect estimate in detail to explain why the number used in meta-analysis is different from the number reported in the study.

About data extraction. According to the Cochrane Handbook for Systematic Reviews of Interventions, we extracted the sample size, the mean and SD of within group change in the intervention group and the control group. Give priority to extracting the data of within group change. If not, extract the mean and SD of the two groups at baseline and after intervention respectively, and calculate the data of within group changes through formula. About data entry and effect estimate. The extracted sample size, mean and SD of the two groups were input into RevMan 5.3 software, and the MD (95%CI) of each study and synthesis could be estimated. In other words, we only extracted within-group data, not between-group data. The between-group data in the manuscript are estimated by software and differ from them in the original study, which are statistically acceptable. For example, in the above table of lean body mass: “Alberga et al., 2012 2.76 0.87 to 4.65” were the between-group data in the original study, and “2.76 0.97 to 4.55” were the between-group data estimated by software. They were certainly different because the results in the original study were calculated using individual data from each patient, while the results in this meta-analysis were calculated using summary data from the original data.

---

## [Editor Report · Decision Letter 1]

31 Jan 2022

The effects of exercise on body composition of prostate cancer patients receiving androgen deprivation therapy: an update systematic review and meta-analysis

PONE-D-21-35409R1

Dear Dr. Zhang,

We’re pleased to inform you that your manuscript has been judged scientifically suitable for publication and will be formally accepted for publication once it meets all outstanding technical requirements.

Kind regards,

Henry Woo

Academic Editor

PLOS ONE

---

## [Editor Report · Acceptance letter]

7 Feb 2022

PONE-D-21-35409R1 

The effects of exercise on body composition of prostate cancer patients receiving androgen deprivation therapy: an update systematic review and meta-analysis 

Dear Dr. Zhang:

I'm pleased to inform you that your manuscript has been deemed suitable for publication in PLOS ONE. Congratulations! Your manuscript is now with our production department. 

Kind regards, 

on behalf of

Prof. Henry Woo 

Academic Editor

PLOS ONE